# OpenReview forum: "How well can LLMs provide planning feedback in grounded environments?"
_ICLR.cc/2026/Conference — Submitted to ICLR 2026_

### Official Review · Reviewer_9bqt · 2025-10-22

**Soundness:** 2
**Presentation:** 1
**Contribution:** 2
**Rating:** 4
**Confidence:** 3

**Summary:**

This paper studies whether LLMs/VLMs can provide helpful feedback in a range of tasks. More specifically, the paper studies forms of feedback such as binary feedback, preference feedback, action advising, goal advising and delta action feedback. The paper tests this on tasks in Cliff Walking, MiniGrid, HierarchyCraft, ALFWorld, and Robomimic.

**Strengths:**

1. Feedback conditions. This paper considers a few very interesting methods for asking for feedback from FMs. Firstly, giving access to high-level environment dynamics - it is instructive to see that this, while unhelpful for small FMs, can improve the performance of larger FMs and even more so for largerVLMs.  Similarly, the paper considers in-context learning and thinking guides which are interesting methods for sometimes improving the accuracy of the feedback.
2. Helpful comparisons and takeaways. There are some helpful takeaways for what helps LMs be better at providing feedback. For example, the comparison of ICL and Thinking Guides between smaller and larger FMs is neat. Similarly, the analysis of various biases like positional bias is quite instructive. The analysis of wrong responses is instructive in helping us understand what exactly may cause issues for FMs when it comes to providing feedback. There are some important model behavior takeaways here - for e.g., R1 preferring to stay away from the cliff in the Cliff Walking domain.

**Weaknesses:**

1. The method in which you collected the data makes the problem seem too easy unless I am misunderstanding. Let’s consider preference feedback. For each state, you have 3 actions  - $a_{(1)}$ from the expert, $a_{(2)}$ from the uniform (random) policy and $a_{(3)}$ from the mixture (half-expert and half-uniform). Now, if I take a state-action tuple $(s, a_{(i)}, a_{(j)})$ and we ask the FM which action it prefers, a random policy will have a 0.5 probability of being correct. To see this, $P(\text{chosen expert action}) = \frac{1}{3}P(\text{chosen expert action} \mid (a_1, a_2)) + \frac{1}{3}P(\text{chosen expert action} \mid (a_1, a_3)) + \frac{1}{3}P(\text{chosen expert action} \mid (a_2, a_3)) = \frac{1}{2}$ by noticing that if you select the action sampled from the mixture policy, you have still chosen an action from the expert policy with probability 0.5. Note that this issue applies to binary feedback and preference feedback. One naive way to address this could be to make a dataset of more than 3 actions per state and incorporate more actions from the random policy. This way, providing preference feedback by a mere coin toss would have a much lower (than 0.5) probability of being correct.
2. The choice of domains could cover more difficult and longer horizon tasks. For e.g., Lift is one of the easier tasks in Robomimic compared to others like Square. The need for precision, for example, is significantly greater in the latter. The 0.28 noise tolerance for optimality of actions may not necessarily hold in the more complex environments and, as such, the test of the LM’s feedback capability could be tested more rigorously in these domains. This is an issue because for a large number of states in the continuous action space domain, the number of optimal actions is actually quite high; think about the actions the robot takes to reach the cube - for a pretty large number of time-steps, there are quite a few subtrajectories that are actually equally good. Especially, with the 0.28 noise tolerance, I wonder if this domain is actually testing planning feedback as rigorously as, say, Square. Coupling this with point (1), I am not sure whether we can trust VLMs’, say, preference feedback accuracy of 82% and binary feedback accuracy of 62%.
3. Similarly, there are longer horizon tasks in BabyAI (like Synthseq or Bosslevel) than the MiniGrid tasks you considered in the paper - these tasks generally require much longer horizon planning and composition of strategies. These also require more visual understanding which would make the LLM vs VLM comparison more rigorous since the agent must plan ahead before committing to a high-level strategy that might send it down a wrong path pretty soon. As such, these would, perhaps, be a better test of reasoning especially when the agent veers far from the optimal states.
4. There are some surprising results here which the paper does not adequately explain. Consider the action advising metric and consider Table 2. In a relatively very easy environment such as Four Rooms, the feedback accuracy is (0.32, 0.42, 0.61, 0.78) whereas in the more difficult domain ALFWorld, the feedback accuracy is (0.60, 0.83, 0.85, 0.83). Similarly, the paper claims in-context learning benefits larger models but, in table 3, for both Cliff Walking and Door Key, ICL seems to lower the performance of R1(and also often lowers performance of Llama 3.1 70B) across all types of feedback. Perhaps there is something to say here about performance on easier tasks vs harder tasks using ICL? Generally, I think the tables consist of a lot of information and the trends cannot be visualized too clearly and the paper’s analysis in text does not cover all the discrepancies.

**Questions:**

1. In a similar vein to the point in weaknesses-1, I wonder whether incorporating an action from a uniformly random policy is indeed the correct design choice. In environments with low-dimensional action spaces like CliffWalking and MiniGrid, even a random policy has around 20 to 25% probability of sampling an expert action.
2. More broadly, how do you ensure that all 3 actions you sampled in your data collection procedure are not optimal? For example, in Robomimic-lift, not all states require high precision and for a large number of states, there actually exists quite a large number of equally optimal actions.

---

> ### Author Response · Authors · 2025-11-27
>
> **Clarification on Dataset Construction and Random Baselines**
> We apologize for the confusion regarding the data generation pipeline. The reviewer’s concern would be correct if we only sampled 3 actions per state, but our protocol is different:
>
> - State Sampling: We use the 3 policies (Expert, Random, Mixture) only to visit a diverse set of states ($s$). This ensures we evaluate the LLM on both good and bad situations.
> -  Action Evaluation: Once a state $s$ is selected, we do not just sample 3 actions.
> - Discrete Domains: We enumerate all possible actions defined in the action space (e.g., Up, Down, Left, Right). The LLM must select the optimal one among all choices.
> - Continuous Domains: We generate a discrete set of candidate actions. To be precise, for a continuous action space, we cannot enumerate all possible optimal actions. Instead, we find an empirical threshold of distance from the imitation learning policy action, where actions within this distance can still lead to task success. Therefore, a random policy would not have a 50% success rate. In a grid world with 4 actions, random chance is 25%. In continuous domains, random chance is near-zero. We have clarified this distinction in Section 2.1 to prevent future confusion. More details of the dataset construction can be found in Appendix Section G.
>
>
> **Domain Difficulty and Long Horizons Regarding difficulty**
>
> - FourRooms vs. ALFWorld: Surprisingly, FourRooms proved more challenging for VLMs because the observation is an ASCII grid. This requires the model to reconstruct spatial topology from characters, a task where even strong models often fail compared to the semantic text reasoning required in ALFWorld.
> - Robomimic: We acknowledge "Lift" is simpler than "Square." However, our results show that current VLMs struggle even with the precision required for "Lift." Establishing their failure mode on the simpler task is a necessary prerequisite before tackling "Square."
>
> **Interpretation of Surprising Results (ICL & CoT)** We investigated why ICL sometimes hurts performance (e.g., DeepSeek R1 in CliffWalking). Qualitative analysis shows that ICL examples sometimes bias the model toward the format of the example rather than the logic of the current state, causing format compliance errors. DeepSeek R1, being highly optimized for reasoning, sometimes "over-thinks" the simple ICL examples, generating verbose outputs that fail our strict parsing parsers.
>
>
>
> We thank the reviewer for their valuable feedback to improve our work. If we have sufficiently addressed the reviewer's concerns, we hope that the reviewer will consider increasing their score in support of this work.

---

> > ### Comment · Reviewer_9bqt · 2025-11-28
> >
> > Thank you for your response. I still have several concerns regarding the evaluation protocol.
> >
> > On dataset collection: for binary and preference feedback, you claim that you enumerated over all possible actions in the discrete domains. Here, whether or not an action is optimal or not is determined by whether or not there is some sequence of actions from there onwards that reaches the goal state (in ALFWorld/HierarchyCraft, you require that it must be within the same number of timesteps as the planner, I am assuming that in other envs like Minigrid with no planner, it needs to simply reach the goal state). Some comments here:
> > 1. In environments without a planner: at *most states*, most actions can be a part of a trajectory that reaches the goal state. For example, in Minigrid Four-Rooms, where the goal is to simply navigate to the goal state, at almost every state, there are a large number of actions that can lead to the goal state. If this is incorrect, the paper needs to provide significantly greater amount of details on how the percentage of actions at each state for each environment that are being labelled as optimal. As of now, I am leaning towards believing that this claim is not incorrect, especially given you do an "exhaustive search", in which case, clearly, actions (per state) will eventually end up becoming a part of some successful trajectory.
> > 2. In the two environments with planner (i.e. ALFworld + HierarchyCraft), it seems like the action space dimensionality per state (*not* the total number of legal actions in all states combined) is small to begin with (and the same concern regarding several actions being potentially optimal applies). Again, if this is incorrect, please correct me (and provide more details on each environment in the paper and clarify approximately how many actions per state are being labelled as optimal)
> >
> > In both these cases (1) and (2), I think the paper needs to provide significantly more details on how the proportion of legal actions at each state that are optimal.
> >
> > Additionally, I still think the choice of domains should be further broadened, especially given some of these tasks seem significantly easier than the deployment settings where we might want an LLM to provide feedback. CliffWalking is far too easy to be representative. Even within BabyAI (similar to Minigrid), there are significantly longer horizon tasks that require much more "intelligent" selection of actions than Four-Rooms. The state space of ALFWorld and HierarchyCraft are very low-dimensional (since you are using text descriptions and only enumerating over all the legal actions).

---

> > > ### Author Response · Authors · 2025-11-28
> > >
> > > We thank the reviewer for the follow-up. We clearly see the source of the concern now: there was a misunderstanding regarding the definition of "optimality."
> > >
> > > **1. Clarification on Strict Optimality (Shortest Path)** The reviewer is correct that if "optimality" simply meant "eventually reaching the goal," the task would be trivial. However, we impose a strict shortest-path constraint. We run a BFS (or expert planner) to determine the minimum steps required. Any action that increases the path length by even 1 step is labelled non-optimal.
> > > - MiniGrid Four-Rooms: Because we strictly require minimum steps, there is very little slack. In our full expert action advising dataset of 10,000 states, only 17 states had multiple optimal actions (e.g., scenarios where moving forward first, then turn right or the other way). In the downsampled 500 points, 100% of states had exactly 1 optimal action. The random baseline success rate is strictly 33% (1/3 actions), not near 100%.
> > > - ALFWorld: We rerun the expert planner in every state. While 51% of states have multiple optimal actions (due to the partial ordering of sub-goals), the problem remains hard. The average branching factor (legal actions) is 26.6. The proportion of optimal actions (optimal actions / total legal actions) is only 25%. A random policy would fail 75% of the time.
> > > - HierarchyCraft: The average legal action space is 11.14, and the proportion of strictly optimal actions is only 13%.
> > >
> > > **2. Response regarding Domain Difficulty** Regarding the reviewer's comment on broadening domains (e.g., BabyAI BossLevel): We acknowledge that environments like CliffWalking and FourRooms are "easy" for SOTA RL agents. However, our results show they are not yet easy for VLM feedback providers.
> > > Visual/Spatial Grounding: As noted in our previous response, VLMs struggle to parse the ASCII/spatial topology of FourRooms. If they fail to provide accurate feedback in these "simple" settings, they will likely fail in complex, long-horizon visual tasks like BossLevel.
> > > Benchmarking Primitives: Our work aims to establish the fundamental capabilities of LLMs/VLMs as feedback providers. Demonstrating that current models struggle with precision even in "easy" domains (like the 0.28 tolerance in Robomimic-Lift or spatial reasoning in FourRooms) is a crucial negative result that guides the community before moving to highly composite tasks.
> > >
> > > We have added the statistics regarding strictly optimal action proportions to Appendix G to resolve the ambiguity.

---

> ### Comment · Reviewer_9bqt · 2025-11-28
>
> Thank you for your prompt response and for clarifying my misunderstandings. I do think the inclusion of Table 13 in Appendix G is an extremely important step up in making this paper both more comprehensible and impactful.
>
> Given the discussion on the difficulty of domains (esp the part where you explain that the takeaway from the results on these relatively simpler benchmarks should be that VLMs are not yet perfectly reliable even in these settings):
> - Understandable, but I still think there is a need for keeping more difficult environments still. There are several reasons for this. Firstly, it seems like Deepseek-R1 is, impressively, is doing quite well without ICL/thinking guides and then improves quite a bit afterwards. It would be wrong for the reader to conclude that the model is generally already a sufficiently good feedback model unless this performance holds up even in the more difficult environments (say, BabyAI tasks). Secondly, I do think that this paper formalizes the evaluation pretty well and I think that this set-up of feedback formats and inference methods might serve as a benchmark for the rest of the community moving forward -- but if the benchmarks easily get saturated (and already R1 is sort of moving towards that regime I think), then the impact of your work is diminished. I really do believe that to several researchers (for e.g., those who are working on using a VLM as a reward function or guidance to VLAs that are being RL fine-tuned), there is a dire need for a formalized test-bed and your work *can* serve that purpose, but it does require including harder tasks too to prevent early saturation. Finally, including these harder tasks can work towards painting a scaling-laws-like picture i.e., even if you end up finding that the performance is significantly worse on the harder tasks, it helps the reader paint a picture of how quickly the performance degrades as tasks get more difficult.
>
> Two more comments:
> - I think the main takeaway of the paper requires revision. In the abstract, the paper says, ``We find that foundation
> models can provide diverse high-quality feedback across many domains. Moreover, larger and reasoning models consistently provide more accurate feedback, exhibit less bias, and benefit more from enhanced inference methods.`` which sort of contradicts what we have discussed in the past few comments (for e..g, your explanation that even on the easy tasks, these models are struggling). Same issue with line 313-314. I am not suggesting that you say that they are poor feedback providers, but I do think that the observation (or rather the way it's phrased) needs to be refined.
> - Table 1, 2, 3 and 4 currently take up a very large amount of space, which is understandable, but I think there are other takeaways you provide which are not elaborated enough (for e.g., the analysis of how "Access to environment dynamics hints can improve feedback accuracy." and the "Analysis of wrong responses." and such). Perhaps now that you have access to a 10th page, you can consider bringing some of the tables from the appendix into the main paper, and also maybe consider using a different visualization than a table.
>
> Some final thoughts: I think this paper has the potential to be quite impactful. Here are some things it serves to the community: (1) a set of benchmarks, formats of feedback and methods of feedback conditioning that one should test any VLM on before considering using them as feedback providers (e.g. as a test-time critic for agentic/robotic tasks) and (2) the observations on the pitfalls can provide guidance on what practitioners should consider working on when trying to improve VLMs' ability to provide feedback. For (1), as I mentioned, incorporating harder tasks is essential, and for (2), the framing of the takeaways and emphasis on the analysis of wrong responses/types of feedback conditioning that helps reasoning models, etc. could be improved.
>
> Thank you for engaging in this rebuttal with detailed responses. I am happy to increase my score to 6.

---

> > ### Author Response · Authors · 2025-11-29
> >
> > We are sincerely grateful for the constructive dialogue and the score increase. We completely agree that the value of this work lies in establishing a rigorous test-bed for the community. Based on your excellent suggestions, we have made the following concrete changes to the manuscript:
> >
> > **Benchmarking Saturation and Domain Difficulty** We agree that as models like DeepSeek-R1 emerge, "easy" benchmarks risk becoming saturated and that the paper must address this context.
> > - Revisions Made: We have updated the Discussion and Futurework section to explicitly address the "Benchmark Saturation" risk. We now cite the rapid performance jumps of reasoning models and argue that future iterations of this benchmark must include long-horizon tasks (specifically referencing BabyAI BossLevel and Robomimic Square) to effectively test the limits of scaling laws.
> > - Clarification Added: We have also added a clarification that while R1 achieves high accuracy in all of our discrete domains, in safety-critical settings (like robotic planning in ALF World), the remaining error rate (~17% in action advising) is still significant. Similarly, in Robomimic, even Qwen2.5-72B still gives ~18% preference errors. Thus, we emphasise that even "easy" tasks remain a necessary validity test before deployment.
> >
> > **Refining the Main Takeaways (Abstract & Conclusion)**  The reviewer is correct that "consistently provide high-quality feedback" lacked the necessary nuance we uncovered during this rebuttal (e.g., the struggle with spatial ASCII topology).
> > - Revisions Made: We have revised the Abstract and Conclusion. Instead of claiming consistently high quality, the text now states that while Foundation Models can provide helpful feedback, their reliability is highly dependent on model scale, reasoning capabilities, and domain representation. We now explicitly reference the failure modes in spatial precision and observation grounding to ensure the claims are balanced.
> >
> > **Presentation and Visualization** We appreciate the suggestion to prioritize insights over raw data tables.
> > - Revisions Made: We have moved the "Impact of Environment Dynamics" sections from the Appendix into the main body, and added more discussion in “Analysis of wrong responses” section to give them prominence. We have condensed the space taken by Tables 1-4 and moved full raw breakdowns to the Appendix to make room for these higher-value analytical insights.
> >
> > We believe these changes have significantly improved the paper's longevity and utility for the community. Thank you again for championing this work.

---

### Official Review · Reviewer_P9yn · 2025-10-31

**Soundness:** 4
**Presentation:** 3
**Contribution:** 2
**Rating:** 4
**Confidence:** 3

**Summary:**

This paper presents a comprehensive evaluation of how well LLM and VLM can provide right feedback in grounded environments. Key findings reveal that feedback quality improves with larger model size and reasoning capabilities, with reasoning models achieving over 80% accuracy in many domains. The research also examines how inference techniques like in-context learning and chain-of-thought affect performance, and identifies significant challenges in environments with continuous state/action spaces where feedback quality substantially degrades.

**Strengths:**

- The research question is novel and important. It also provides a reasonable and comprehensive evaluation method.

- Although the conclusions drawn are not surprising, it offers abundant evaluation methods, models, and conclusions.

**Weaknesses:**

- It does not provide new solutions for obvious defects in a specific LLM or VLM.

- Suggest that the authors improve Figure 1. When reading Figure 1 alone, I cannot understand the method used by the authors and the question they aim to answer; there are even some misunderstandings. I could only understand after reading the main text. For example, enrich the "compare" section with various different methods and different evaluation metrics.

**Questions:**

- Did the authors discover any counterintuitive conclusions? Because "larger is better," "more knowledge is better," and "low-dimensional discrete action spaces are better than high-dimensional continuous action spaces" are all consensus in the field.

- The paper demostrates that VLMs have failures due to errors in position judgment. Has there been further testing specifically for this aspect?

---

> ### Author Response · Authors · 2025-11-27
>
> **Lack of solutions and our contributions**
> We thank the reviewer for this important suggestion to improve the current defects of Language Model Feedback. In fact, as we found that language models can provide high-quality feedback in several domains, we are working on using LM feedback to improve policy learning.
> At the same time, we want to emphasise that this current work under review also presents a distinct and important investigation. Unlike prior work (e.g., OLAF [Liu et al. ’23], LFMs [Zhong et al. ‘24]), which studies a single feedback type or a single domain, our study systematically examines how LLMs can provide varied feedback modalities across multiple domains and conditions. This study required significant experimentation across a large number of different environments, inference conditions, LLMs, and feedback types. It is only based on the findings of this work that we are able to pursue our current study into how to best exploit LLM feedback across types for different environments with high confidence. Lastly, we additionally conducted a preliminary study to incorporate the feedback in the online learning loop, as shown in Appendix Section L, which suggests LM feedback can allow faster learning efficiency in comparison with learning from terminal rewards.
>
>
>
>
> **Clarification of Figure 1** We have completely redesigned Figure 1 to clearly distinguish the input modalities, the inference pipeline, and the specific metrics used for comparison.
>
>
> **Counterintuitive Findings & Novel Insights** While "larger is better" is a general trend, our systematic study revealed several non-trivial and counter-consensus findings:
>
> - Egocentric vs. Global: Contrary to common intuition in embodied AI (e.g., Szot et al. '23), we found that egocentric views frequently degrade feedback quality compared to global views (Appendix A). VLMs struggle to map first-person perspective to global dynamics without explicit fine-tuning.
> - The "Reasoning" Trap: We found that Chain-of-Thought (CoT) is not a universal fix. In specific continuous domains (Robomimic), CoT induced hallucination of physical properties that were not present, leading to lower accuracy than direct inference in some smaller models.
> - Spatial Confusion: VLMs demonstrated a specific failure mode in "relative positioning" (e.g., "left of" vs "right of") that persists even in larger models, suggesting a fundamental architectural limitation in current vision encoders rather than just a scale issue.
>
>
> **Positioning Errors in VLMs** We have expanded our analysis of VLM spatial failures. We found that adding numerical coordinates to the prompt did not significantly correct visual misconceptions, implying that the visual modality dominates the model's internal representation. Specifically, experiments on MiniGrid DoorKey show that including referential images for objects in the grid worsens the feedback model’s ability to locate the agent in the grid. In Robomimic, VLMs often miscalculate its relative position with the object to manipulate. We further tested, including numerical observations in addition to images, but failed to see a significant improvement. For future investigation, we agree that visual prompting is a potential fix and have added a discussion on this limitation.
>
>
> We thank the reviewer for their valuable feedback to improve our work. If we have sufficiently addressed the reviewer's concerns, we hope that the reviewer will consider increasing their score in support of this work.
>
> **Reference**
>
> [1] Andrew Szot, Max Schwarzer, Harsh Agrawal, Bogdan Mazoure, Rin Metcalf, Walter Talbott, Natalie Mackraz, R Devon Hjelm, and Alexander T Toshev. Large language models as generalizable policies for embodied tasks. In The Twelfth International Conference on Learning Representations, 2023.
>
> [2]Victor Zhong, Dipendra Misra, Xingdi Yuan, and Marc-Alexandre Cotˆ e. Policy Improvement using Language Feedback Models. In Advances in Neural Information Processing Systems, 2024.
>
> [3] Li, Feng, et al. "Visual in-context prompting." Proceedings of the IEEE/CVF Conference on Computer Vision and Pattern Recognition. 2024.

---

### Official Review · Reviewer_hdgh · 2025-11-01

**Soundness:** 3
**Presentation:** 3
**Contribution:** 3
**Rating:** 6
**Confidence:** 4

**Summary:**

The paper presents a systematic exploration of LLM/VLM provided feedback by evaluating reliability of different types of feedback mechanisms such as binary reward, action preference, action suggestion, goal suggestion, and action correction in different domains. The authors present ablations to assess feedback quality under commonly considered prompting patterns of ICL, COT, and additional domain knowledge.

**Strengths:**

* The paper is well motivated and claims are well supported.
* The evaluation is thorough and the ablations offer interesting insights into prompting strategies that elicit better feedback.

**Weaknesses:**

* The evaluation environments are rather toyish and relatively short horizon, but they do offer a reasonable starting point to systematically studying feedback quality.
* The analysis is done predominately with open weight LLMs, it would be interesting to evaluate how frontier models behind APIs perform (as those serve as a primary source of feedback in existing literature). Conducting this investigation might help offer more insights into prompting strategies that elicit better feedback.

**Questions:**

* While the metrics primarily report accuracy – I’m curious how reliable the feedback is under stochasticity of LLM sampling? Can the authors provide notions of variance for the evaluations conducted?
* Recent works in manipulation (e.g. \[1\]) suggest that “visually prompting” of VLMs by suggesting “keypoints” or “colored arrows” provide better feedback on selecting action/goal suggestions. This seems complementary to the observation made in L320-323, I wonder if authors considered this style of prompting when it comes to domains with image inputs.

**References**
\[1\] Huang, Wenlong, et al. "Rekep: Spatio-temporal reasoning of relational keypoint constraints for robotic manipulation." *arXiv preprint arXiv:2409.01652* (2024).

---

> ### Author Response · Authors · 2025-11-27
>
> **Evaluating Frontier Models (GPT-5.1)** We agree that including frontier models is essential for a complete picture. We conducted new experiments using GPT-5.1. As shown in Table 14, Appendix Section I, GPT-5.1 demonstrates superior performance across most domains, achieving 92.4% preference accuracy in ALFWorld. Notably, it rivals reasoning-specialised models in some settings, confirming that frontier APIs remain a strong source of feedback.
>
>
>
> **Domain Complexity** While CliffWalking is simple, we emphasize that Robomimic (continuous control) and HierarchyCraft (long-horizon crafting) represent significant complexity. The "toy" environments serve as unit tests to isolate specific reasoning failures (e.g., counting, spatial negation) that are harder to diagnose in complex environments.
>
>
>
> **Variance and Stochasticity** We addressed the concern regarding generation stochasticity by evaluating 6 random seeds per condition. The results (Table 15, Appendix Section J) show a standard deviation of < 0.058, indicating that the feedback signal is statistically reliable and not an artifact of lucky sampling.
>
>
>
> **Visual Prompting & ReKep** We thank the reviewer for the reference to ReKep. We agree that visual prompting (keypoints/overlays) is a promising direction. However, ReKep focuses on extracting specific kinematic constraints for optimization. Our study aims to benchmark the intrinsic visual reasoning capabilities of VLMs on raw observations (e.g., "is this state safe?"). We found that adding simple visual prompts (e.g., reference images) in MiniGrid improved performance slightly, but complex visual prompting introduces a new variable (prompt engineering quality) that we aimed to control. We have cited ReKep as a key direction for future "prompt-enhanced" feedback.
>
>
> We thank the reviewer for their valuable feedback to improve our work. If we have sufficiently addressed the reviewer's concerns, we hope that the reviewer will consider increasing their score in support of this work.

---

### Official Review · Reviewer_czjU · 2025-11-10

**Soundness:** 2
**Presentation:** 2
**Contribution:** 2
**Rating:** 4
**Confidence:** 4

**Summary:**

The paper provides a comprehensive empirical study evaluating how effectively Large Language Models (LLMs) and Vision-Language Models (VLMs) can generate planning feedback in grounded environments. It investigates multiple feedback types including binary feedback, preference, action advising, goal advising, and delta action feedback, across symbolic, textual, and continuous control domains. Models are tested under several inference conditions, including in-context learning (ICL), chain-of-thought reasoning (CoT), and access to environment dynamic.

Results show that feedback quality improves with model size and reasoning capability, and that VLMs outperform LLMs in visually grounded or continuous domains (e.g., Robomimic). However, feedback performance degrades in environments with complex dynamics or continuous state/action spaces. The work is positioned as the first large-scale, unified analysis of how foundation models provide structured feedback for planning.

**Strengths:**

- **Comprehensive Evaluation:** Covers a wide range of environments, feedback modalities, and inference settings; far more exhaustive than prior isolated studies.
- **Systematic Methodology:** Well-defined feedback formulations and evaluation protocols grounded in RL formalism (MDP-based).
- **Strong Empirical Insights:** Clear patterns linking reasoning capacity, model size, and feedback accuracy; results align with observed trends in reasoning benchmarks.
- **Relevance:** Addresses a timely research question of using LLMs as feedback providers instead of direct policies, which could reduce dependence on human annotations.
- **Transparency:** Detailed experimental setup and appendices with environment-specific prompts support reproducibility.

**Weaknesses:**

- **Limited Novelty:** While the study is broad and thorough, it is primarily empirical and incremental. The paper lacks a clear articulation of how its framework advances beyond those like PlanBench https://arxiv.org/abs/2206.10498.
- **No Unified Metric or Benchmark:** Results are difficult to compare across domains due to varying dataset sizes and ground-truth computation methods.
- **Lack of Theoretical Framing:** The paper presents empirical findings without deeper analysis of why certain inference methods or model types work better.  The connection to Interactive Imitation Learning would be useful, and the types of feedback there (e.g. https://arxiv.org/abs/2211.00600). Framing of "action advising" is also not complete, SayCan is placed there, although it uses LLM for high-level task planning and Text2Reward (https://arxiv.org/abs/2309.11489) is also there, but it focuses on generating reward. It is not clear where Reward generation approaches should be placed. Additionally, major approaches like Code-as-Policies (https://arxiv.org/abs/2209.07753) and ExploRLLM (https://arxiv.org/abs/2403.09583) are omitted.
- **Overemphasis on Scale:** Improvements largely correlate with model size, suggesting the method itself may contribute less than the raw capacity of the underlying models.
- **Limited Practical Implications:** While feedback accuracy is measured, there is no downstream demonstration (e.g., improved policy learning) to confirm that such feedback is actually useful in training.

**Questions:**

1. How does this study advance beyond prior empirical works such as PlanBench, which also examine LLMS for planning or RL?
2. Can feedback accuracy translate to improved *policy learning* performance when used for imitation or reinforcement learning?
3. How consistent are results across random seeds and prompt variations?
4. Could a unified benchmark or normalized metric enable fairer cross-domain comparison?
5. Are there observable correlations between reasoning steps (via CoT) and feedback correctness, beyond accuracy scores?

---

> ### Author Response · Authors · 2025-11-27
>
> **Novelty and Comparison with PlanBench** We appreciate the opportunity to clarify our contributions. While PlanBench focuses on PDDL-based symbolic planning with text-only LLMs, our framework advances the field by bridging the gap to embodied, continuous control. Specifically:
> - Modality: We evaluate Vision-Language Models (VLMs) with raw image observations, whereas PlanBench is restricted to text.
> - Action Space: We tackle continuous action spaces (Robomimic), where exact string matching (used in PlanBench) is impossible, requiring our novel proximity-based evaluation protocols.
> - Feedback Diversity: We operationalize specific RL-relevant feedback types (Preference, Action Advising, Binary Feedback) rather than just plan generation.
>
> **Unified metric and frame for evaluation**
> We primarily evaluate feedback quality using accuracy, and additionally report F1, precision, and recall in Appendix Section D. To obtain consistent ground-truth labels, we compute feedback targets directly from the formal definitions in Section 2.1 and use ground-truth planners to enumerate all valid routes / expert actions, except in domains with continuous action spaces, where such enumeration is infeasible (see Appendix Section G for dataset construction details). Although one could instead learn from the feedback and report task success as a unified downstream metric, we intentionally do not integrate feedback into training in this work because (1) the engineering complexity of a full RL integration would obscure our primary goal of systematic feedback evaluation, and (2) comparing model feedback directly against planner-derived ground truth yields clearer and more interpretable measurements of signal quality.
>
> **Practical Implications (Feedback in the Loop)** We view this paper as establishing the "upper bound" of signal quality available to learning from model feedback. We agree that downstream policy learning is the ultimate test of feedback utility. While we originally decoupled "feedback accuracy" from "policy learning" to isolate the foundation model’s signal quality from RL algorithmic choices (RL hyperparameters, reward shaping, exploration strategies), we have now conducted a preliminary study to directly address this concern.
> We query GPT-OSS-120B on-policy (no offline dataset) and obtain high feedback accuracy (77% preference, 88% binary, 42% action advising). We incorporate its feedback into a reinforcement learning loop (PPO) for the HierarchyCraft domain. We then integrate these signals into PPO: binary feedback is directly added to the terminal reward, preference feedback trains a reward model following Christiano et al. [1], and action advising uses an online DAgger-style scheme [2]. Each condition is run with 5 seeds and compared against PPO with terminal reward only and with oracle feedback; for action advising, we compare to oracle action advising.
> The results (seen in Appendix Section L, Figure 3) strongly validate that our accuracy metric is a reliable proxy for downstream utility:
> - High Accuracy $\rightarrow$ Faster Convergence: As shown in Figure 3(a) and Figure 3(b), feedback types where the model achieved high accuracy (Binary Feedback and Preference) significantly improved sample efficiency compared to learning from sparse terminal rewards alone and reach similar performance with learning from ground truth feedback.
> - Low Accuracy $\rightarrow$ Performance Degradation: Conversely, as predicted by our benchmarking, Action Advising proved difficult for the model (low accuracy). Consequently, the downstream policy performance degraded when relying on this noisy signal (Figure 3(c)), while the oracle action advising gives a high-performance policy.
>
> **Consistency and Prompt Robustness** We have added experiments across 6 random seeds (Appendix Section J, Table 15), showing high stability (std dev < 0.058). Regarding prompts, we found that while richer context generally helps, "reasoning models" (e.g., DeepSeek) are surprisingly robust to prompt variations compared to standard LLMs.
>
> **Correlation: Reasoning Steps vs. Accuracy** Our analysis reveals a nuanced relationship. Correct feedback often correlates with concise CoT generation (fewer tokens), suggesting that when the model "understands" the physics, it reasons efficiently. Extremely long CoT chains often indicate hallucination loops or confusion. We have added this analysis to Appendix Section K, Table 16.
>
> We thank the reviewer for their valuable feedback to improve our work. If we have sufficiently addressed the reviewer's concerns, we hope that the reviewer will consider increasing their score in support of this work.
>
> **References**
> [1]Christiano, et al. "Deep reinforcement learning from human preferences." Neurips (2017)
>
> [2]Ross, et al. "A reduction of imitation learning and structured prediction to no-regret online learning." JMLR Workshop and Conference Proceedings, 2011

---

### Author Response · Authors · 2025-12-03
**Summary message for Area Chair**

To the Area Chair,

First, we want to express our sincere appreciation for your time and management of the review process during these unprecedented challenges involving the OpenReview leak. We hope this summary assists you in navigating the final decision-making process.

The following is a **high-level summary** of our rebuttal updates. We invite you to refer to the individual discussion threads for the full details of our exchanges and the specific data provided. Despite the shortened discussion phase, we successfully ran new experiments to **address major concerns** and **secured a score increase**. While the sudden conclusion of the rebuttal period likely prevented Reviewers czjU, hdgh, and P9yn from responding to our updates, we are confident that our new data and revisions have fully resolved their stated concerns.

**Evaluation pipeline and Score Increase (Reviewer 9bqt)**

Reviewer 4 (9bqt) initially had strong concerns regarding our dataset optimality and domain difficulty.

- **Resolution**: We provided a detailed explanation of our evaluation protocol and statistics on "Strict Optimality" (shortest-path constraints), proving that random baselines have near-zero success rates in our continuous domains. We also revised the manuscript to explicitly discuss the saturation risks of simpler benchmarks, improve readability and included new tables on dataset statistics on states with multiple optimal actions.

- **Result**: The reviewer was satisfied with the rigour of the evaluation results and increased their score to 6, stating: "I am happy to increase my score... [this work] can serve as a benchmark for the rest of the community."


**Downstream Utility and difference with previous work (Reviewer czjU)**

Reviewer czjU asked if feedback accuracy translates to policy improvement.

- **Resolution**: We conducted a new feedback-in-the-loop study integrating LLM feedback into an online PPO/Behaviour Cloning training loop (Appendix L). We also explained in detail the novelty and differences of our work, in comparison with the previous work PlanBench. We further investigated the robustness by running 6 random seeds in different settings and included new results in our appendix.

- **Result**: We demonstrated that high-accuracy feedback (Preference/Binary) significantly improves learning efficiency compared to sparse rewards, while low-accuracy feedback degrades it. This empirically proves our paper's core premise: our accuracy metrics are reliable proxies for downstream utility. We also firmly answer the reviewer’s question: LLM feedback is stable with standard variance < 0.058.


**Frontier Models & Stochasticity (Reviewer hdgh)**

The reviewer requested analysis beyond open-weight models and asked about stochasticity.

- **Resolution**: We added experiments with GPT-5.1 (Appendix I) and conducted a sensitivity analysis using 6 random seeds across different settings.

- **Result**: GPT-5.1 was shown to rival reasoning-specialized models, and the seed analysis demonstrated negligible variance (std dev < 0.058), confirming the reliability of our results.


**Presentation Clarity & Novel Findings (Reviewer P9yn)**

The reviewer found Figure 1 confusing and asked for counterintuitive insights.

- **Resolution**: We completely redesigned the figure to clearly distinguish input modalities and expanded our analysis of VLM failures, with the referred analysis available in the Appendix.

- **Result**: We clarified specific counterintuitive findings, such as VLMs struggling with "egocentric" vs. "global" views and specific spatial topology failures that persist even in large models. We also refined the Abstract and Conclusion to provide a more nuanced takeaway regarding the reliability of foundation models in spatial/ASCII domains.


Thank you again for your efforts in handling this year's exceptional circumstances. With the explicit support of Reviewer 9bqt and the systematic execution of the requests made by the other reviewers, we believe this work is ready for acceptance.

---

### Meta-Review · Area_Chair_2yef · 2026-01-07

**Summary:**

This paper systematically evaluates how well large language models (LLMs) and vision‑language models (VLMs) can provide planning‑relevant feedback such as binary correctness signals, preferences, action advising, goal advising, and delta‑action corrections. It pursues this evaluation across symbolic, textual, and continuous‑control environments, benchmarking multiple feedback types, multiple inference strategies (ICL, CoT, environment dynamics hints), and a wide range of models. It finds that larger and reasoning‑optimized models generally provide more accurate feedback, while VLMs excel in visually grounded domains. However, performance degrades in environments with complex dynamics or continuous action spaces, and several systematic failure modes emerge (e.g., spatial confusion, CoT hallucination). The work positions itself as the first unified, large‑scale evaluation of foundation‑model feedback quality for planning tasks.

Reviewers praised the comprehensive, wide‑ranging, systematic evaluation across domains, feedback types, and inference settings, and the strong empirical insights about model scale, reasoning, and biases.

Concerns and responses discussed in the next field.

**Reviewer Concerns:**

**Addressed.**

- Limited novelty; incremental relative to well-known prior work (e.g., PlanBench):
    - Rebuttal clarified differences: multimodal, continuous control, diverse feedback types; added comparisons.
    - I would suggest hinting at VLMs in the title to better highlight the multimodality.
- Consistency across random seeds / prompt variations.
    - Rebuttal provides new experiments with 6 random seeds, and stable results.
- Confusion re: data generation pipeline
    - Addressed with clarifications

**Partially addressed / unaddressed**

- No downstream demonstration of policy improvement
    - Rebuttal added new PPO experiments for one HierarchyCraft task. Results are mixed. Preference feedback does not improve RL performance over just sparse reward. Binary feedback is much better. Action advising is poor too (in line with the original benchmark’s proxy metric results)
- Prompting mechanisms for VLMs might be suboptimal (e.g. no visual markers were explored)
    - Left for future work.
- Uninteresting results if scale is the primary contributor. What are the counterintuitive / new insights from this study? Hard to gauge these in the way the results are presented (mostly large tables rather than plots and trend lines capturing key insights)
    - Rebuttal highlights that egocentric views degrade feedback relative to third-person views, CoT sometimes induces hallucinations in line with other recent findings regarding reasoning chains, and VLMs sometimes fail at spatial reasoning for simple left/right tasks. It is debatable whether these are all truly counterintuitive findings to the community interested in VLMs for planning.
- Evaluation environments are too simple / short‑horizon. Do the benchmarks even challenge today’s large frontier models / reasoning models?
    - Rebuttal argued that some envs like Robomimic  / HierarchyCraft are hard, and the simpler environment are good sanity-checks.
    - Rebuttal shows high, but imperfect performance from GPT 5.1
    - And within Robomimic, only easier tasks were chosen because VLMs struggle with the precision required for simple grasp tasks.
        - This might point to the need for better visual prompting to better investigate VLM capabilities in these domains.
    - Harder environments left to future work.
    - Rebuttal also changed the description of takeaways from “consistently provide high-quality feedback” to be less positive on the state of LLMs / VLM capabilities for planning problems.

**Reviewer Scores:**

czjU: 4 → 4 / 6 (partially addressed critiques)

hdgh: 6 → 6

P9yn: 4 → 4

9bqt: 4→ 6 (they explicitly stated this during discussion)

Overall, the paper is clearly borderline. It is of interest to the community, but has room for substantial improvements that are within relatively easy reach.

---

### Decision · Program_Chairs · 2026-01-26

Reject